# Use of Water from Petroleum Production in Colombia for Soil Irrigation as a Sustainable Strategy Adapted from the Oman Desert

**Angie Tatiana Ortega-Ramírez** [1,*] , **Ivonne Angulo-De Castro** [1] , **Nubia Liliana Becerra** [1] ,
**Juan Camilo Gómez Caipa** [1] and **Victor Alexei Huerta-Quiñones** [2,*]

1 Management, Environment and Sustainability Research Group, Chemical and Environmental Engineering Department, Universidad de América, Bogota 110311, Colombia
2 Petroleum, Natural Gas and Petrochemical Engineering Faculty, Universidad Nacional de Ingeniería, Lima 15333, Peru
* Correspondence: angie.ortega@profesores.uamerica.edu.co (A.T.O.-R.); ahuerta@fip.uni.edu.pe (V.A.H.-Q.)

**Abstract:** Production water represents a major sustainability challenge for oil and gas industries, which is why various strategies have emerged globally to encourage its reuse in proactive projects. One of the most recognized strategies has been developed in the Omani desert where artificial wetlands were designed to physically and biologically treat production water. The present study proposes to adapt this strategy to a Colombian context to further study the sustainability of production water reuse. The methodology of this study consists of three phases: evaluate in-field production water through the characterization of irrigation water, identify and prioritize the variables of said strategy, and propose an adequate soil irrigation strategy for a Colombian field. An expert matrix and multicriteria analysis are used to identify the level of interaction among the variables presented, according to the criteria of experts in the field of sustainable production water management. Water quality was ranked by the experts as the most important single variable. On a scale of 1–5, the variables with the highest level of interaction (2.8) are water quality and the type of treatment seedlings, and the variables with the lowest level of interaction (1.3) are additional water treatment systems and final water use. These results have led to the formation of a hierarchy of variables according to impact, which has been used to create a triple bottom line diagram and demonstrate the feasibility of implementing the Omani strategy in Colombia.

**Keywords:** expert matrix; multicriteria analysis; oil; production water; sustainability; water treatment

## 1. Introduction

The petroleum industry requires high volumes of water for its operations, generating constant challenges in water management and final disposal. There is thus a need to minimize water use during operations to maximize the efficiency of the water–petroleum relationship and, in this way, guarantee more financial viability, making hydrocarbon extraction projects as profitable as possible.

Produced water represents a major challenge for the petroleum industry. Because of this, alternatives for the reuse of produced water have been developed around the world, which seek to promote a more sustainable and environmentally friendly industry. In 2000, in the USA, about 210 million barrels of water were consumed per day (oil and gas industry), and petroleum companies reported spending $40 billion on water management [1]. In 2004, Schlumberger reported the production of three barrels of water for every barrel of petroleum in the world [2]. In Colombia, Ecopetrol reported in 2016 the production of twelve barrels of water for each barrel of petroleum, together with the consumption of 56.23 million $m^3$ of water during the various activities of the petroleum industry [3]. Ecopetrol has developed new strategies to reduce the pressure on water resources caused

by the excessive volume of water collected from sources adjacent to the fields operated by the company, and then spilled, resulting in the reuse of 21.9 million m$^3$ of water in the second quarter of 2020, ceasing to capture and spill this volume.

Ecopetrol reported that for the second quarter of 2020, 789,000 m$^3$ of production water was reused in agricultural and livestock activities in the Agro-Energy Sustainability Area of the Castilla Field [4].

The risks associated with the improper management of production water are due to its high content of fats and oils, heavy metals (strontium, cadmium, barium, lead, chromium, and mercury), anions (sulfates, carbonates, and bicarbonates), gases (oxygen, chlorine, and hydrogen sulfide), and microorganisms, especially sulfate-reducing bacteria [2]. These substances significantly impact the health of people living in the areas surrounding the petroleum fields, altering the composition of the soil and surrounding water sources. The most common cause of illness or death from exposure to petroleum components is aspiration pneumonia [5]. In addition, petroleum contaminants can be deposited in the soil or ingested by aquatic organisms. These contaminated organisms can then enter the food chain and be consumed by people, damaging their health and increasing the malnutrition rates of the local population, especially children and those involved in the fishing industry [6].

To become a more sustainable industry, actors in the petroleum sector have sought to reuse production water in various industrial and agricultural processes or activities [7]. Though these processes do encompass other economic sectors, the largest volume of production water is currently reused in the operations of the petroleum industry [2].

Different strategies for reusing production water have emerged around the world, especially in the agricultural sector, through crop irrigation, the recovery of dry areas, and grazing as livestock feed [8]. In these strategies, outcomes are influenced by several parameters, such as the physicochemical properties of the water, soil type, germination, morphology, growth rate, and the biochemical aspects of the vegetables [9]. Agricultural projects that have used production-water-reuse strategies include:

- Parts of the Libyan Desert, located 3000 km southeast of Benghazi, were recovered through the development of artificial wetlands, where native *Pharagmites australis* were planted. This project led to the reuse of 377–503 barrels of produced water per [7].
- Production water used as irrigation water in hydroponic crop development led to the reuse of approximately 50,000 barrels of water per day [10].
- The reuse of 12,500 barrels of production water was associated with methane extraction in the process of recovering 40.5 hectares of arid land and as crop irrigation water [8].
- In Moscow, researchers investigated how livestock wastewater irrigation affects the quality and agricultural potential of soil through germination experiments conducted on radish seeds irrigated with livestock wastewater [11].

One of the most recognized strategies was developed in the Oman desert, where there is a high production of water associated with petroleum extraction: approximately 800,000 m$^3$ produced per day [12]. The strategy implemented in the Omani desert includes the design of wetlands that allow for the natural physical and biological treatment of the produced water. Due to the success of the project in the desert of Oman, which is considered one of the most efficient strategies for treating and reusing production water, the methodology of the project was used for this present study with adaptations made for the geographical, operational, environmental, and social conditions of Colombia, specifically the location of the Castilla Field in the Eastern Llanos Basin.

The Eastern Llanos Basin has a total area of 200,000 km$^2$ and includes the departments of Arauca, Vichada, Casanare, and Meta [13]. The basin presents high hydrocarbon potential, containing 1506 wells drilled throughout 81 minor fields, two giant fields, and two major fields, Apiay and Castilla [14]. The basin also shows significant agricultural production, especially in the cultivation of rice, African palm, banana, corn, soy, cotton, and sugar cane. The reuse of production water in crop irrigation activities would guarantee a strong interaction between two of the most significant economic sectors in the studied area.

The Eastern Llanos Basin has numerous water sources that could be affected by the release of production water due to its chemical content. Consequently, Resolution 631 of 2015 requires specific physical, chemical, and/or biological treatments for production water, based on deposit geology, for its subsequent discharge into the environment [15]. The chemical composition of production water varies depending mainly on the lithological composition of the producing well and the chemical additives used in the drilling and completion fluid. The concentration of each constituent depends on factors such as pressure or temperature. Production water from the Castilla Field is characterized by a high content of anions such as sulfates, carbonates, and bicarbonates as well as heavy metals, fats, and oils [16].

It is estimated that for every 170,000 barrels of petroleum extracted in the Castilla Field, four million barrels of water are produced [17]. This high production of both hydrocarbons and water represents a challenge for Ecopetrol and has encouraged the development of efficient alternatives for the use of the production water, reducing impact on water resources and/or ecosystems. One of these alternatives, developed by Siemens, uses processes to purify production water for its later discharge or reinjection into the Eastern Llanos Basin [18].

Internationally, important strategies have been developed to reuse production water. One of the most outstanding was developed by the company Petroleum Development Oman along with the German company Bauer, which in 2011 built The Oasis of the Petroleum Industry, named after the idea for the largest wetland for the treatment of industrial water in the world [19]. The treatment plant, located in the Omani desert, is considered one of the most innovative and environmentally friendly projects and has won major global awards [20]. The concept of the treatment plant is based on the development of artificial wetlands composed of plants, substrates, water, and microorganisms that, through biological, chemical, and physical processes, eliminate various pollutants present in production water, ensuring better water quality that facilitates its reuse [21]. The use of the wetland has two main functions: the elimination of residual hydrocarbons and the reduction of water volume through the high evaporation and transpiration rate of the plants [22].

The implementation of a strategy for the treatment of production water in the Eastern Llanos Basin, similar to the one designed in the Omani desert, would generate great benefits to the petroleum sector as well as to the communities surrounding the hydrocarbon extraction fields. However, it is first necessary to evaluate the environmental conditions of the basin and the characteristics of the production water in the Castilla Field, which differs in composition from the water obtained from the production wells of the Nimr Field in the Omani desert. Because of this, the present research has developed a detailed analysis of the parameters described above along with the identification of critical variables that must be considered in wetlands design in order to assure the social, environmental, and economic sustainability of the implementation of a strategy adapted from the Omani project to reuse water from oil production for soil irrigation.

## 2. Materials and Methods

The methodology was based on five steps: diagnosis of the current management of production water in a Colombian field; generalities of the soil irrigation water strategies in Nimr, Oman; identification and prioritization of variables of the Oman strategy; identification of variables to promote the environmental, economic, and social sustainability of the strategy adapted to a Colombian field; development of a strategy for the sustainable use of production water for irrigation of soils in a Colombian field. The following is a methodological table describing the process (Figure 1):

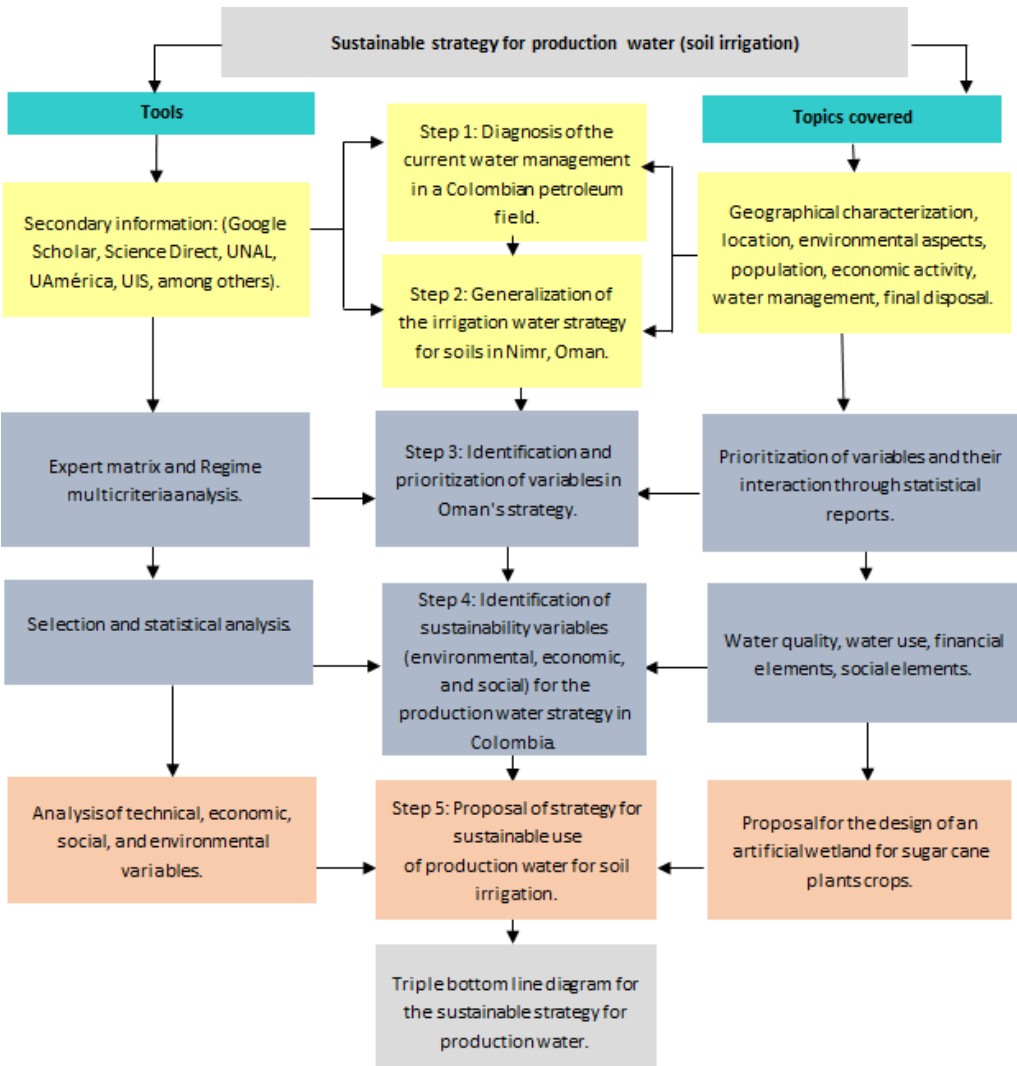

**Figure 1.** This is a figure. Schemes follow the same formatting. Source: Author's elaboration.

Throughout this research, different tools were used to achieve a robust analysis of the information gathered in order to evaluate the viability of a sustainable methodology for the reuse of production water in crop irrigation activities in Colombia. The following is a description of the main tools and/or methodologies used during the five steps previously described:

*2.1. Expert Matrix*

A widely used validation methodology in which specific topics are investigated in a quantitative and/or qualitative manner through consultation with a group of experts, who contribute their experience, background, and specific knowledge from a real context that is not found in historical data, and is difficult to include in mathematical models; this represents a fundamental point for recognizing aspects with a high level of importance, providing reliability, hierarchy, and validity to the evaluated criteria [23]. For this research, a sample size of 10 engineers was selected after a detailed evaluation of the professional profile of a group of petroleum engineers with extensive experience in the area of deposits, production, and treatment of production water.

*2.2. Multicriteria Analysis*

A support tool widely used for describing, selecting, hierarchizing, and identifying variables of a problem or process, recognizing the importance of each variable while

acknowledging that not all of them affect the final result or process in the same way, identifying the links between variables, and, finally, proposing a rational solution through choosing the best alternative [24]. There are several types of multicriteria analyses, and their application depends on the type of information to be analyzed and the expected result. For this research, a Regime-type multicriteria analysis was used to identify qualitative and quantitative information by assigning a value to each of the variables to be able to measure the effect and importance of each parameter with regards to the different scenarios being evaluated [25]. A bibliographic review and background analysis revealed that the variables involved in the development of a production-water-reuse strategy are: final water use (FWU), treatment seedlings in the artificial wetland (TS), climatic conditions (CC), additional water treatment systems (ATS), and water quality (WC).

### 2.3. Statistical Analysis

A process of preparing and analyzing qualitative and quantitative variables to interpret them, through their proper classification, correct association, and convenient transformation into relative figures such as coefficients, averages, rates, and ratios, among others [26]. This analysis is complemented by graphs that evaluate the behavior of one variable with another on which it depends directly or indirectly. For the statistical analysis in this article, the Regime method was used in conjunction with expert judgement. This is a discrete multicriteria method [27] that provides technical support for decision making.

### 2.4. Triple Bottom Line Diagram

A methodology designed by John Elkington in 1998 to evaluate sustainability by proposing a new framework to measure performance which goes beyond the traditional measures of benefits, return on investment, and shareholder value, and includes environmental and social dimensions [28]. The idea is to meet the demands of the stakeholders in a project by understanding the economic, social, and environmental results [29].

## 3. Results

### 3.1. Characterization of the Strategy of Nimr, Oman

The strategy developed in the Omani desert assures a more sustainable use of production water by using it as irrigation water for crops, after undergoing chemical and biological treatment through artificial wetlands. These wetlands were designed based on the characteristics of the climate, soil, and production water from the petroleum fields of Nimr, Oman.

The Sultanate of Oman is located in the southeast of the Arabian Peninsula, east of Saudi Arabia and Yemen. To the north and south, it is bordered by the Oman and Arabian seas, part of which is separated by the United Arab Emirates. Oman has a generally hot and dry climate in its interior and a hot and humid climate on its coasts, where the world's largest artificial wetland is located [30].

The Nimr Water Treatment Plant in the Omani desert can treat 115,000 m$^3$ of produced water per day through surface flow within 350 hectares of artificial wetlands, which are complemented by evaporation ponds covering approximately 500 hectares to reduce the volume and salt content of the production water. Water entering the treatment plant is considered brackish as it contains total dissolved solids (TDS) of approximately 7000 mg/L with an average oil content of 400 mg/L, of which 260 bbl. per day is recovered at the front of the system through the use of hydrocutions and skimmers. Finally, the remaining hydrocarbons in the treated water are biologically degraded within the artificial wetlands, generating an influent with an oil content of less than 0.1 mg/L and salinity of approximately 10,000 mg/L of TDS [31].

These artificial wetlands use an aquatic plant called *Phragmites australis*, also known as the common reed, which can grow in wetlands and resist different levels of salinity. Because it adapts to extreme climates and supports salinity, both in the water and in the soil where it is planted, *P. australis* can be found around the world. During the planting

process of around 240 hectares of the wetland, approximately 1.2 million seedlings of this species were planted, separated by one square meter each [32]. *P. australis* was evaluated in the present study for use in the Eastern Llanos Basin, taking into account the conditions of the water from the Castilla Field.

### 3.2. Identification and Prioritization of Variables

To assess the feasibility of replicating the methodology of the Oman project in the Eastern Llanos Basin, it was first necessary to identify critical variables in the design of artificial wetlands. Selected experts were interviewed to prioritize the identified variables of designing an artificial wetland for production water treatment. Initially, the experts were asked to assign a value of 1 to 5 to each variable depending on its importance to the design of a wetland to treat production water. In this way, the variable with the highest score represents the most important variable to prioritize for research. One of the experts assigned the same value to more than two variables, for which his answers were invalid for this question. After aggregating the values, as shown in Figure 2, final water use emerged as the most important variable, followed by water quality, treatment seedling, climatic conditions, and, finally, additional water treatment systems.

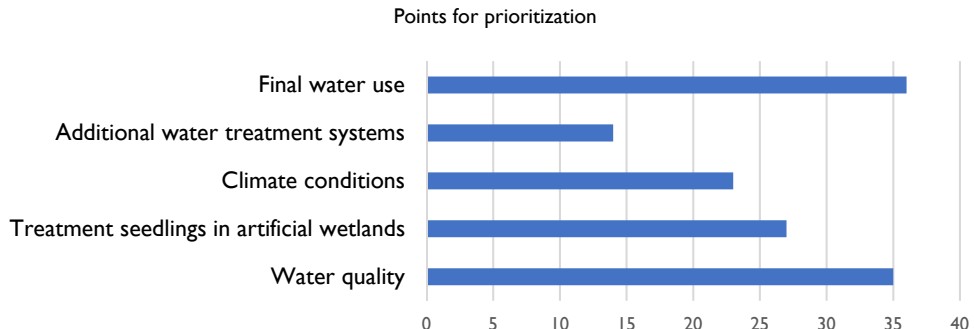

**Figure 2.** Order of prioritization results. Source: Author's elaboration.

In order to prioritize variables, this study used the Regime Method, a multicriteria technique that assigns a value to each variable through an average of the results offered by the experts for each parameter. The average is then used to represent one single value for a given set of quantities. Further, it can be difficult to analyze all the data, and hence the average of the data is taken to represent all the data and, in this way, determine the level of importance of each variable according to the average value of the quantities given by the experts. These results are presented below in Table 1.

**Table 1.** Value for each variable.

| Variable | Average |
|---|---|
| Water quality | 3.9 |
| Treatment seedlings in artificial wetland | 3.0 |
| Climate conditions | 2.6 |
| Additional water treatment systems | 1.6 |
| Final water use | 4.0 |

Source: Author's elaboration.

Subsequently, the experts were asked to evaluate the interaction between the variables as being either null, weak, or high. The results of the Regime method show that the variables of water quality and treatment seedlings in artificial wetlands had a high level of interaction. Water quality and final water use also displayed a high level of interaction. On the other hand, additional water treatment systems and final water use were the variables that presented the lowest level of interaction. These results can be seen in Table 2, whose maximum theoretical value is 3.

**Table 2.** Value for each variable interaction.

| Variable | Treatment Seedlings in Artificial Wetland | Climatic Conditions | Additional Water Treatment Systems | Final Water Use |
|---|---|---|---|---|
| Water quality | 2.8 | 1.9 | 2.7 | 2.8 |
| Treatment seedlings in artificial wetland | | 2.1 | 2.0 | 2.3 |
| Climatic conditions | | | 1.3 | 1.8 |
| Additional water treatment systems | | | | 2.7 |
| Final water use | | | | |

Source: Author's elaboration.

Finally, the Regime model enabled a graphic analysis of the results obtained with the expert matrix. Therefore, the levels of importance of the variables (Figure 3) and each interaction (Figure 4) were identified graphically.

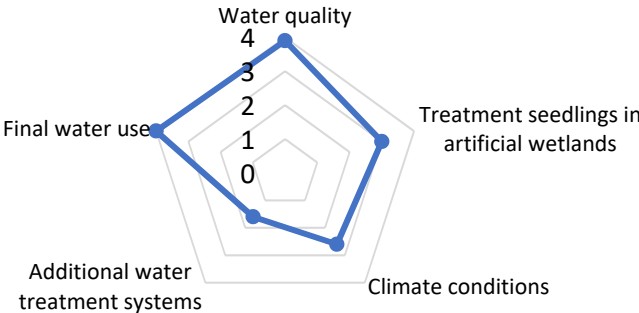

**Figure 3.** Hierarchy of variables. Source: Author's elaboration.

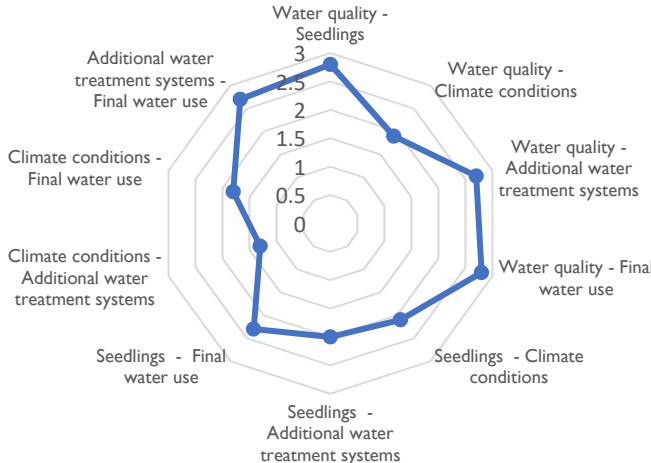

**Figure 4.** Interaction of variables. Source: Author's elaboration.

*3.3. Characterization of Nimr and Colombia's Production Water*

Based on the results obtained with the expert matrix, the quality of the production water was identified as one of the most important variables. Therefore, a comparison of the main characteristics was made between the water produced in the Nimr petroleum fields in the Omani desert and the water produced in the Castilla Field in the Eastern Llanos Basin in Colombia. The comparison data, compiled in Table 3, indicate a lower concentration of sodium, chlorine, and total sulfur as sulfate ions ($SO_4$) in the Colombian

case. On the other hand, water from the Oman case presented a lower concentration of boron, calcium, magnesium, and $HCO_3$. Consequently, because it contained a higher concentration of calcium and magnesium, the production water from the Castilla Field presented a higher facility for the formation of incrustations or corrosion in the equipment along with a reduction in the efficiency of herbicides due to the precipitation of insoluble salts [33]. The irrigation of this water in Colombian fields may affect the properties of the Colombian soil, deteriorating characteristics such as fertility, organic matter content, drainage, and structural stability.

**Table 3.** Comparative table of the properties of Omani water vs. Colombian water.

| Parameter | Oman | Colombia | Colombia vs. Oman |
|---|---|---|---|
| Sodium (mg/L) | 2470 | 1490 | Lesser |
| Chlorine (mg/L) | 3201 | 2570 | Lesser |
| Boron (mg/L) | 4.59 | 8.95 | Greater |
| $HCO_3$ (mg/L) | 189 | 378 | Greater |
| Calcium (mg/L) | 62 | 429 | Greater |
| Magnesium (mg/L) | 22 | 72.30 | Greater |
| Total Sulfur as $SO_4$ (mg/L) | 283 | 8.15 | Lesser |
| $Ca/SO_4$ (mg/L) | 0.22 | 52.64 | Greater |
| Ca/Mg (mg/L) | 2.82 | 5.93 | Greater |
| Na/Cl (mg/L) | 0.77 | 0.58 | Lesser |
| Cl/Br (mg/L) | 697.39 | 287 | Lesser |

Source: Author's elaboration.

In order to increase the viability of the project, it is essential to comply with Colombian regulations for irrigation water, described in Decree 1076 of 2015 [34] through the issuance of the Sole Regulatory Decree of the Environmental and Sustainable Development Sector. Table 4 compares various parameters of the water produced in the Colombian field with respect to national regulations, identifying that the values of several parameters, such as conductivity, total phenols, chlorides, and sodium, are outside of the limits of these regulations for agricultural processes. However, the values of several parameters such as pH, sulfates, and iron are within the Colombian regulations. The consequences of the implementation of this resource for the agricultural communities surrounding the sector must be considered; the use of produced water would represent a great economic benefit for promoting new crop irrigation technologies and sustainable social growth in the area.

**Table 4.** Comparative table of study production water properties vs. Colombian regulations.

| Parameter | Unit of Measurement | Colombian Case | Colombian Normative Limit | Suitable/Unsuitable |
|---|---|---|---|---|
| pH at 25 °C | pH units | 7.22 | 6.0–9.0 | Suitable |
| Conductivity | µS/cm | 8850 | 1500 | Unsuitable |
| Total phenols | mg/L | 8.04 | 1.5 | Unsuitable |
| Chlorides | mg Cl/L | 3000 | 300 | Unsuitable |
| Sulfates | mg $SO_4$/L | 8.15 | 500 | Suitable |
| Iron | mg Fe/L | 2.62 | 5.0 | Suitable |
| Sodium | mg Na/L | 1490 | 200 | Unsuitable |

Source: Author's elaboration.

### 3.4. Biological Characterization of the Omani and Colombian Plants

To effectively design artificial wetlands for the biological treatment of production water, various external factors that intervene in the process must be evaluated. These factors include location, temperature, wind, vegetation, and the correlation between evaporation and precipitation in the wetlands.

One of the most important factors is the vegetation to be implanted in the artificial wetlands. Its optimal growth depends directly or indirectly on the other factors described; the vegetation of a wetland is a critical part of important processes, from the degradation

of hydrocarbons dissolved in water to the transportation of oxygen to low depths. When the vegetation dies, it then becomes a substrate for the growth of a microbial film.

Due to the difference in climate and production water between the desert of Nimr in Oman and the Castilla Field in Colombia, a different type of plant should be used in the artificial wetlands, one that is correctly adapted to the climatic conditions and the location of the Colombian field under study. For this reason, the sugar cane plant (*Saccharum officinarum* L.) was used. This plant has thick and fibrous stems that can grow between 3–5 m high with an approximate thickness of 2–5 cm and a high content of sucrose, processed to obtain sugar [35].

Table 5 presents a comparison between the common reed and the sugar cane of main biological characteristics. The sugar cane presents a longer leaf length and a limitation of the soil pH, which must be between 5.5–7.8, reducing the cultivable zones using this type of plant. In the same way, the characteristics of the stem and its height are differentiated, since local sugar cane presents a thick and fibrous stem that can reach a height of five meters, while the common reed is composed of a long and woody stem that can reach a height of 6 m with a diameter of 2 cm; it is observed to be thinner than sugar cane.

**Table 5.** Comparative table of biological properties between the Colombian and Omani plants.

| Plant Properties | Sugar Cane (Colombia) | Common Reed (Oman) |
|---|---|---|
| Scientific name | *Saccharum officinarum* L. | *Phragmites australis* |
| Kingdom | Vegetable | Vegetable |
| Family | *Poaceae* | *Gramineae* |
| Tribe | *Andropogoneas* | *Arundinae* |
| Genus | *Saccharum* | *Phragmites* |
| Species | *Spontaneum* and *Robustum* | *Australis* and *Chrusathus* |
| Relative humidity | Medium or high | Medium or high |
| pH | 5.5–7.8 | Unlimited |
| Stem height | 5 m | 6 m |
| Stem diameter | 2–5 cm | 2 cm |
| Leaf length | 30–60 cm | 20–45 cm |
| Leaf width | 1–5 cm | 1–5 cm |
| Inflorescence length | 20–60 cm | 15–50 cm |

Source: Author's elaboration.

### 3.5. Project Sustainability Assessment

Based on the triple bottom line (TBL) methodology, the economic, social, and environmental aspects involved must be assessed to ensure balance and meet the demands of the various stakeholders in the project. As a consolidation of the study, the conclusive aspects are presented, starting from the variables evaluated in the economic, social, and environmental factors involved in the implementation of the proposed sustainable strategy:

- Economic: The economic variables evaluated in the project are related to technical-financial aspects such as different techniques to ensure water quality. These evaluate additional treatment systems and the optimal seedlings for this project. Likewise, costs for the implementation of an artificial wetland must be established to increase the financial viability of the project [36]. This viability must consider capital costs associated with the required infrastructure (adaptation of land, etc.) and the operating costs based on the cost per volume of reused water (supplies, labor for artificial wetland and crop water quality monitoring activities, maintenance and harvesting of the plant, etc.).

- Social Factor: During the development of a project, it is important to evaluate the potential positive and/or negative impact on the local communities. Because of this, the social variables are related to the Colombian regulations for agricultural activities and the different groups with a direct or indirect interest in the project [36]. These interests can influence the social acceptance of the project, which highlights the importance of communicating clear and truthful information to the various interest

groups about the implications of its development. This ensures that they have the necessary elements to adopt a position of acceptance or rejection and guarantees the right of access to information and citizen participation according to the mechanisms established in national regulations. In relation to the right of access to information, The Interstate Technology and Regulatory Council suggests that the information to be communicated must include at the least: potential risks to the community, possible impacts on the production capacity of the land, and whether there are nearby sensitive ecosystems [37].

- Environmental Factor: The environmental variables consist of those characteristics that can be evaluated for the development of an artificial wetland that generates the least possible impact on the environment. Likewise, it is necessary to evaluate the features and biological requirements of the sugar cane [36].

As a result of the analysis of the factors involved, Figure 5 illustrates the diagram of the triple bottom line, where the direct connections among the technical–financial, social, and environmental variables are identified.

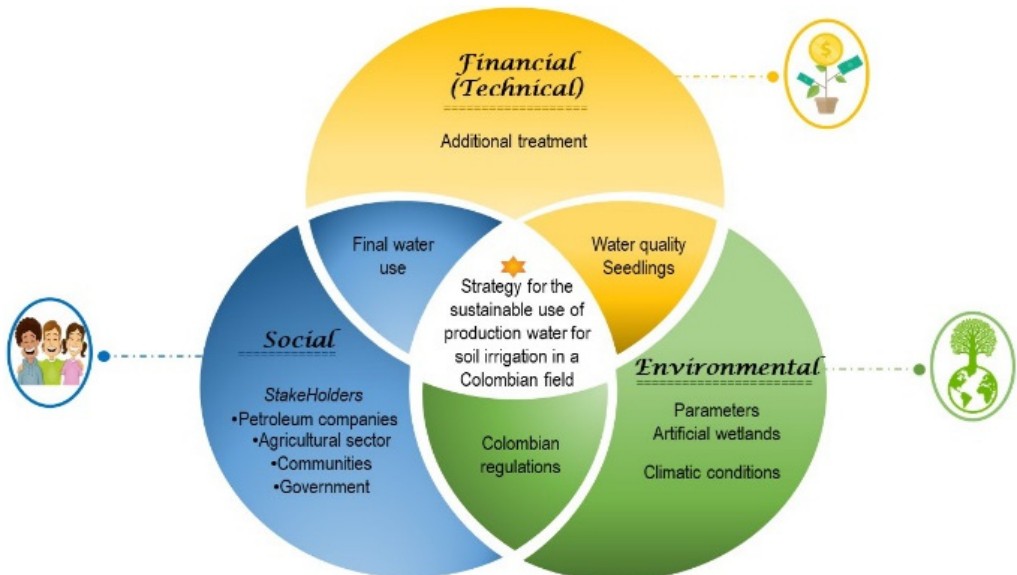

**Figure 5.** Triple bottom line diagram. Source: Author's elaboration.

The triple bottom line diagram demonstrates a first approach to the implementation of Oman's strategy in Colombia in the use of produced water in irrigation activities. The approach seeks to assure the viability and sustainability of the project, focused on a green economy that promotes income creation while at the same time maintaining the balance among the variables of profit, community benefit, and environmental impact prevention [38].

## 4. Conclusions

The implementation of a sustainable strategy in Colombia for the reuse of production water adopted from Oman is feasible and represents a significant advance in the sustainability of the petroleum industry in Colombia, considering its scope. However, during the development of the project, it is necessary to evaluate different economic, social, and environmental aspects to meet the demands of the stakeholders. This requires the joint analysis of the different parameters that are presented under the same phenomenon, such as regulations, water quality, affected communities, and climatic conditions, among others. However, according to the results obtained in this research, one of the variables with the greatest degree of importance when implementing a sustainable strategy for the reuse of production water is water quality because it affects the degree of complexity of the treatments required for production water to be used as irrigation water without altering the physical and chemical properties of the soil. The water present in the Eastern Llanos Basin

presents a lower concentration of sodium, chlorine, and total sulfur as sulfate ions ($SO_4$) than the water in Oman. Because of this, its treatment will be easier in Colombia; however, as it contains a higher concentration of calcium and magnesium, the water presents a greater facility for the formation of incrustations or corrosion in the equipment and a reduction in the efficiency of herbicides due to the precipitation of insoluble salts. Additionally, this research identified that the sugar cane presents a thick and fibrous stem that can reach a height of five m, while the common reed is composed of a long and woody stem that can reach a height of six m with a diameter of two cm. For this reason, biological waste from sugar cane crops is recommended as biomass material for the production of bio-oil, bioethanol, biogas, and other derivatives, taking into account the quality of the treated production water. Further research should be conducted to understand potential toxic effects due to the composition of the treated water and the implications of its irrigation. Additionally, further research on the operating conditions of the wetlands would enrich the body of knowledge on this topic. Finally, more knowledge could be developed around the implementation of this technology, delving into issues such as the bioconversion of sugar cane or the biodegradation of recalcitrant compounds present in treated water.

**Author Contributions:** Conceptualization, A.T.O.-R. and I.A.-D.C.; methodology, A.T.O.-R. and I.A.-D.C.; validation, V.A.H.-Q. and J.C.G.C.; formal analysis, A.T.O.-R. and I.A.-D.C.; investigation, N.L.B. and A.T.O.-R.; resources, I.A.-D.C.; data curation, N.L.B.; writing—original draft preparation, I.A.-D.C., N.L.B. and A.T.O.-R.; writing—review and editing, V.A.H.-Q. and N.L.B.; visualization, V.A.H.-Q. and N.L.B.; supervision, V.A.H.-Q. and J.C.G.C. All authors have read and agreed to the published version of the manuscript.

**Funding:** This research received no external funding.

**Institutional Review Board Statement:** Not applicable.

**Informed Consent Statement:** Not applicable.

**Data Availability Statement:** Not applicable.

**Conflicts of Interest:** The authors declare no conflict of interest.

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
