# Peer review of "Use of Water from Petroleum Production in Colombia for Soil Irrigation as a Sustainable Strategy Adapted from the Oman Desert"

_sustainability, doi:10.3390/su142214892_

Round 1

Reviewer 1 Report

The research presented meets the requirements of the journal.

Author Response

good morning

I send the letter with response to each of the comments.

Thank you

Reviewer 2 Report

The paper is well written and provides a desktop study comparing a constructed wetland in Oman to a proposed constructed wetland in Colombia. The irrigation waters are chemically different and the Colombian water appears to be a Ca/Mg carbonate based water which, as stated by the authors, will need some alternative pre-treatment design.

Oman uses "reeds" in their constructed wetlands, but then "cane" is proposed for Colombia? I can see why, and Table 5 provides the authors summary, which is helpful. For clarity, maybe describe both reed and cane differences, but in your comparison just state that  "alternative constructed wetland vegetation was investigated due to the water quality differences"....or something like that. For example, you mention a "different cane" was used in Colombia but it was actually a "reed" used in Oman, and it makes it a bit confusing.

The TBL discussion is not strong, or maybe just not clear. The Expert Matrix provides a great foundation for defining important issues but the TBL does not really provide the "minimum variable"....which is usually the economic bottom line. Probably difficult to expand on without knowing treatment scenarios and/or additonal treatment costs.

Overall, an interesting paper that is well structured and provides a desktop comparison of a working constructed wetland in Oman, and an approach to potentially transferring that success to a Colombian constructed wetland.

Author Response

good morning

We send an attached file with the response to your comments and suggestions

Thank you

Reviewer 3 Report

This is an interesting and practical work on serving waste water recycled from petroleum industry to irrigate soil lands. The paper describes the concept and application of the issue very well and I may suggest this valuable work for publication after some minor modifications as follows:

1. Please review the abstract. It seems you have mentioned three phases instead of five.

2. Please furnish the abstract with numerical results. Numbers say more than words.

3. I may suggest the esteemed authors to enrich the literature review with more references relevant to the use of different types of waste water in land irrigation.

4. Please use suitable subsection for methodology section. It is somehow misleading.

5. Do you have any suggestion for future works?

Good luck,

Author Response

Good morning

We attach the document with the changes made

Thank you
